# IDENTIFYING LATENT CAUSAL CONTENT FOR MULTI-SOURCE DOMAIN ADAPTATION

## ABSTRACT

Multi-source domain adaptation (MSDA) learns to predict the labels in target domain data, under the setting that data from multiple source domains are labelled and data from the target domain are unlabelled. Most methods for this task focus on learning invariant representations across domains. However, their success relies heavily on the assumption that the label distribution remains consistent across domains, which may not hold in general real-world problems. In this paper, we propose a new and more flexible assumption, termed *latent covariate shift*, where a latent content variable $\mathbf{z}_c$ and a latent style variable $\mathbf{z}_s$ are introduced in the generative process, with the marginal distribution of $\mathbf{z}_c$ changing across domains and the conditional distribution of the label given $\mathbf{z}_c$ remaining invariant across domains. We show that although (completely) identifying the proposed latent causal model is challenging, the latent content variable can be identified up to scaling by using its dependence with labels from source domains, together with the identifiability conditions of nonlinear ICA. This motivates us to propose a novel method for MSDA, which learns the invariant label distribution conditional on the latent content variable, instead of learning invariant representations. Empirical evaluation on simulation and real data demonstrates the effectiveness of the proposed method.

## 1 INTRODUCTION

Traditional machine learning requires the training and testing data to be independent and identically distributed (Vapnik, 1999). This strict assumption may not be fulfilled in various potential real-world applications. For example, in medical applications, it is common to seek to train a model on patients from a few hospitals and generalize it to a new hospital (Zech et al., 2018). In this case, it is often reasonable to consider that the distributions of data from training hospitals are different from the new hospital (Koh et al., 2021). Domain adaptation is a promising research area to handle such problems. In this work, we focus on multi-source DA (MSDA) settings where source domain data are collected from multiple domains. Formally, let $\mathbf{x}$ denote the input, *e.g.* image, $\mathbf{y}$ denote the labels in source and target domains, and $D$ denote the domain index. We observe labeled data pairs $(\mathbf{x}^{\mathcal{S}}, \mathbf{y}^{\mathcal{S}})$ from the multiple joint distributions $p(\mathbf{x}, \mathbf{y}|D = 1), ..., p(\mathbf{x}, \mathbf{y}|D = m), ..., p(\mathbf{x}, \mathbf{y}|D = M)$ in source domains, and unlabeled input data samples $\mathbf{x}^{\mathcal{T}}$ from the joint distribution $p(\mathbf{x}, \mathbf{y}|D_{\mathcal{T}})$ in the target domain. The training phase of MSDA is to use the sets of $(\mathbf{x}^{\mathcal{S}}, \mathbf{y}^{\mathcal{S}})$ and $\mathbf{x}^{\mathcal{T}}$, to train a predictor so that it can provide a satisfactory estimation for $\mathbf{y}^{\mathcal{T}}$ in the target domain. The key for MSDA is to understand how the joint distribution $p_D(\mathbf{x}, \mathbf{y})$ change across all different source and target domains.

Most early methods assume that the change of the joint distribution results from *Covariate Shift* (Huang et al., 2006; Bickel et al., 2007; Sugiyama et al., 2007; Wen et al., 2014), *e.g.*, $p_D(\mathbf{x}, \mathbf{y}) = p_D(\mathbf{y}|\mathbf{x})p_D(\mathbf{x})$, as depicted by Figure 1(a). This setting assumes that $p_D(\mathbf{x})$ changes across domains, while the conditional distribution $p_D(\mathbf{y}|\mathbf{x})$ is invariant across domains. Such assumption may not always hold for some real applications, *e.g.*, image classification. For example, the assumption of invariant $p_D(\mathbf{y}|\mathbf{x})$ implies that $p_D(\mathbf{y})$ should change as $p_D(\mathbf{x})$ changes. However, we can easily change style information (*e.g.*, hue, view) in the images to change $p_D(\mathbf{x})$ and keep $p_D(\mathbf{y})$ unchanged, which is common in classification but violates the assumption.

In contrast to covariate shift, most current works consider *Conditional Shift* as depicted by Figure 1(b). It assumes that the conditional $p_D(\mathbf{x}|\mathbf{y})$ changes while $p_D(\mathbf{y})$ is invariant across domains

(Zhang et al., 2013; 2015; Schölkopf et al., 2012; Stojanov et al., 2021; Peng et al., 2019). This situation motivates a popular class of methods focusing on learning invariant representations across domains to approach the latent content variable $\mathbf{z}_c$ in Figure 1(b) (Ganin et al., 2016; Zhao et al., 2018; Saito et al., 2018; Mancini et al., 2018; Yang et al., 2020; Wang et al., 2020; Li et al., 2021; Stojanov et al., 2021). However, the label distribution $p_D(\mathbf{y})$ may change across domains in many real application scenarios (Tachet des Combes et al., 2020; Lipton et al., 2018; Zhang et al., 2013), for which learning invariant representations may be resulting in degenerating performance. In theory, there exists an upper bound on the performance of learning invariant representations when label distribution changes across domains (Zhao et al., 2019).

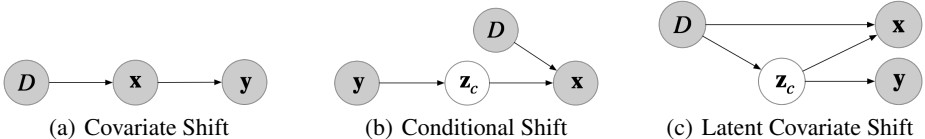

(a) Covariate Shift  (b) Conditional Shift  (c) Latent Covariate Shift

Figure 1: The illustration of three different assumptions for MSDA. (a) Covariate Shift: $p_D(\mathbf{x})$ changes across domains, while $p_D(\mathbf{y}|\mathbf{x})$ is invariant across domains. (b) Conditional Shift: $p_D(\mathbf{y})$ is invariant, while $p_D(\mathbf{x}|\mathbf{y})$ changes across domains. (c) Latent Covariate Shift: $p_D(\mathbf{z}_c)$ changes across domains while $p_D(\mathbf{y}|\mathbf{z}_c)$ is invariant across domains.

In some real-world scenarios, the label distribution $p_D(\mathbf{y})$ and the corresponding $p_D(\mathbf{z}_c)$ may vary across domains. For example, in image classification, $\mathbf{z}_c$ represents the essential visual semantic features for discriminating classes (*e.g.*, fur texture or color for animal classification) may vary across domains (*e.g.*, geographical location) but are also stably associated to specific domains. In such scenarios, if all domain variant features are excluded from $\mathbf{z}_c$, $\mathbf{z}_c$ will be non-informative for deciding $\mathbf{y}$. On the other hand, since such varying features are associated with and covariant with the domain, they can be usable and useful in MSDA. To handle this scenario, we propose a new assumption, *Latent Covariate Shift (LCS)*, as depicted by Figure 1(c). Unlike the conditional shift, LCS assumes that there is a latent content variable $\mathbf{z}_c$, whose distribution $p_D(\mathbf{z}_c)$ changes across domains. Meanwhile the label conditional distribution $p_D(\mathbf{y}|\mathbf{z}_c)$ is invariant. By combing these assumptions, LCS enables the label distribution to change across domains.

To understand more deeply and handle LCS, we propose a latent causal model to formulate the data and label generating process, by introducing the latent style variable $\mathbf{z}_s$ to complement $\mathbf{z}_c$ as depicted in Figure 2. To analyse the identifiability of the proposed causal model, we also introduce latent noise variables $\mathbf{n}_c$ and $\mathbf{n}_s$, which represent some unmeasured factors influencing $\mathbf{z}_c$ and $\mathbf{z}_s$, respectively. As a result, we can leverage recent progress in the identifiability result of nonlinear ICA (Hyvarinen et al., 2019; Khemakhem et al., 2020) to analyse the identifiability of the proposed latent causal model. We show that although completely identifying the proposed latent causal model is often not possible without further assumptions due to transitivity in latent space, partially identifying the latent content variables $\mathbf{z}_c$ up to scaling is tractable, by integrating the identifiability result of nonlinear ICA with the dependence between $\mathbf{n}_c$ and $\mathbf{y}$. This motivates us to propose a novel method to learn the invariant conditional distribution $p_D(\mathbf{y}|\mathbf{z}_c)$ for LCS, instead of learning invariant representations. Relying on the guaranteed identifiability on $\mathbf{z}_c$, the proposed method provides a principled way to ensure that the covariant $\mathbf{z}_c$ can be identified on the target domain data, and the learned predictor $p_D(\mathbf{y}|\mathbf{z}_c)$ can generalize to the target domain. Empirical evaluation on synthetic and real data demonstrates the effectiveness of the proposed method, compared with state-of-the-art methods.

Overall, our main contributions can be summarized as follows: (i) Differ from the commonly-used Conditional Shift as shown in Figure 1 (b), which assume label distribution to be the same across domains, we propose a new problem setting, *latent covariate shift*, as shown in Figure 1 (c). (ii) We propose a latent causal model for *latent covariate shift*. Leveraging the existing identifiability results of nonlinear ICA, we provide an analysis about the identifiability of the proposed latent causal graph, which provides guarantee for identifying the latent causal content variable $\mathbf{z}_c$. (iii) Under the identifiability result, we design a new method for domain adaptation, and empirically evaluate the proposed method on simulation and real data.

## 2 RELATED WORK

**Learning invariant representations**. Due to the limitation of covariate shift in image data, most current works for domain adaptation consider the conditional shift, which learns invariant representations across domains (Ganin et al., 2016; Zhao et al., 2018; Saito et al., 2018; Mancini et al., 2018; Yang et al., 2020; Wang et al., 2020; Li et al., 2021; Wang et al., 2022b; Zhao et al., 2021). Such invariant representations can be obtained by applying suitable linear or nonlinear transformations on the input data. The key of these methods is how to enforce the invariance of the learned representations. For example, the invariance can be enforced by maximum classifier discrepancy (Saito et al., 2018), or by a domain discriminator for adversarial training (Ganin et al., 2016; Zhao et al., 2018; 2021), or by moment matching (Peng et al., 2019), or by relation alignment loss (Wang et al., 2020), or by pseudo labeling (Wang et al., 2022b). However, all these methods require label distribution to be invariant across domains. As a result, when label distribution is varying across domains, they may perform well only in the overlapping areas among label distributions in different domains, while facing with challenges in the non-overlapping areas. To overcome this, recent progress focuses on learning invariant representations conditional on the label across domains (Gong et al., 2016; Ghifary et al., 2016; Tachet des Combes et al., 2020). One of the challenges in these methods is that the labels in the target domain is unavailable. More importantly, these methods do not guarantee that the learnt representations to be consistent with the true relevant information for predicting in the target domain, thus there is no principled way to guarantee that the learned predictor can generalize to the target domain.

**Learning invariant conditional distribution** $p_D(\mathbf{y}|\mathbf{z}_c)$. There exist few of works exploring the invariant conditional distribution $p_D(\mathbf{y}|\mathbf{z}_c)$ for domain adaptation (Kull & Flach, 2014; Bouvier et al., 2019). Differ from these two works, the proposed method provides the identifiability of $\mathbf{z}_c$, so that the learned $p_D(\mathbf{y}|\mathbf{z}_c)$ in this work can generalize to the target domain in a principled way. Besides, in the context of out-of-distribution generalization, some recent works explore learning invariant conditional distribution $p_D(\mathbf{y}|\mathbf{z}_c)$ (Arjovsky et al., 2019; Sun et al., 2021; Liu et al., 2021; Lu et al., 2021). For example, Arjovsky et al. (2019) imposes learn the optimal invariant predictor across domains from the viewpoint of an intimate link between invariance and causation, while the proposed method directly explores conditional invariance given the proposed latent causal model. Sun et al. (2021) mainly focus on single domain, while the proposed method consider multiple domains. The proposed method is also different from the work in Liu et al. (2021) in that the former assume the latent content variable caused by the style variable, while the latter depends on a confounder to model the causal relation between the latent content variable and the style variable. Unlike the work in Lu et al. (2021) that the label is treated as a variable causing the other latent variables, the proposed method assumes that the label have no child nodes.

**Causality for Domain Generalization** It has been shown that there is closely the connection between causality and generalization (Peters et al., 2016). Motivated by this, most of current methods leverage to introduce new methods in various applications, *e.g.*, domain generalization, (Mahajan et al., 2021; Christiansen et al., 2021; Wang et al., 2022a),text classification (Veitch et al., 2021), Out-of-Distribution Generalization Ahuja et al. (2021). Perhaps the closest to our problem setting is domain generalization, where one can not 'see' input data $\mathbf{x}$. In general, because one can not 'see' input data $\mathbf{x}$ for domain generalization, obtaining identifiability result in the setting of domain generalization is generally not possible. In contrast, this work provides the identifiability result, providing a principled way to guarantee that the learned predictor can generalize to the target domain.

## 3 THE PROPOSED LATENT CAUSAL MODEL FOR LATENT COVARIATE SHIFT

We introduce a latent causal model to formulate features and label generative process to handle LCS as depicted by Figure 2. It introduces the observed domain variable $D$ to denote in which specific domain data are collected. $\mathbf{n}_c$ and $\mathbf{n}_s$ represent some unmeasured factors corresponding to latent content noise and latent style noise, respectively. $\mathbf{n}_c$ and $\mathbf{n}_s$ influence the latent content variable $\mathbf{z}_c$ and the latent style variable $\mathbf{z}_s$, respectively. Generally speaking, $\mathbf{z}_c$ and $\mathbf{z}_s$ should be dependent given the domain variable $D$. Here we consider that $\mathbf{z}_c$ causes $\mathbf{z}_s$, to model the correlation between $\mathbf{z}_s$ and $\mathbf{y}$. In the proposed latent causal model, $p_D(\mathbf{z}_c)$ change across domains while $p_D(\mathbf{y}|\mathbf{z}_c)$ is invariant across domains, which meets the basic assumption in the proposed latent covariate shift.

In the following, we discuss two key causal assumptions, which also highlight the novelty of the proposed latent causal model.

$\mathbf{z}_c$ **causes y:** Previous works consider the causal relation between $\mathbf{x}$ and $\mathbf{y}$ as $\mathbf{y} \rightarrow \mathbf{x}$ (Gong et al., 2016; Stojanov et al., 2019; Li et al., 2018), while we employ $\mathbf{z}_c \rightarrow \mathbf{y}$. We argue that these two cases are not contradictory since the labels $\mathbf{y}$ in these two cases represent two different physical meanings. To understand this point, let $\hat{\mathbf{y}}$ replace $\mathbf{y}$ in the first case (i.e., $\hat{\mathbf{y}} \rightarrow \mathbf{x}$) to distinguish from $\mathbf{y}$ in the second case (i.e., $\mathbf{z}_c \rightarrow \mathbf{y}$). For the first case, consider the following generative process of images. A label should be first sampled, e.g., $\hat{\mathbf{y}}$, then one

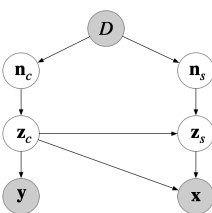

Figure 2: The proposed latent causal model.

may determine content information regarding to the label $\hat{\mathbf{y}}$, and finally generate a image, which is a reasonable assumption in many real application scenarios. In the proposed latent causal model, $\mathbf{n}_c$ play a role to replace $\hat{\mathbf{y}}$ and causes the content variable $\mathbf{z}_c$. For the second case, $\mathbf{z}_c \rightarrow \mathbf{y}$, it formulates the process that experts extract content information from given images and then provide reasonable labels according to their domain knowledge. This assumption has been employed by some recent works (Mahajan et al., 2021; Liu et al., 2021; Sun et al., 2021). Particularly, these two different labels, $\hat{\mathbf{y}}$ and $\mathbf{y}$, has been simultaneously considered in Mahajan et al. (2021).

$\mathbf{z}_c$ **causes** $\mathbf{z}_s$**:** Here we consider having the object essence $\mathbf{z}_c$ first, from which a latent style $\mathbf{z}_s$ springs to render $\mathbf{x}$. The correlation between $\mathbf{y}$ and $\mathbf{z}_s$ can be seen as a spurious correlation that should not contribute to predicting $\mathbf{y}$. We here employ $\mathbf{z}_c$ as a confounding factor of both $\mathbf{y}$ and $\mathbf{z}_s$ to model the spurious correlation. The rationality of this assumption can be further verified from the viewpoint of the converse. In particular, if we assume that $\mathbf{z}_s$ causes $\mathbf{z}_c$, all high-level information in input data $\mathbf{x}$, $\mathbf{z}_s$ and $\mathbf{z}_c$ would be causally related to the label $\mathbf{y}$, which can not model the spurious correlation and is obviously unreasonable. Therefore, assuming $\mathbf{z}_c \rightarrow \mathbf{z}_s$ is more persuasive and consistent with previous works (Gong et al., 2016; Stojanov et al., 2019; Mahajan et al., 2021). One recent work in Sun et al. (2021) leverages an additional variable as a confounding factor that influences both the content variable $\mathbf{z}_c$ and the style variable $\mathbf{z}_s$ to model their relation. Interestingly, the identifiability result in Sun et al. (2021) does not depend on the confounding factor. As a result, the confounding factor can be incorporated into the domain index, which is equivalent to the case where $\mathbf{z}_c$ and $\mathbf{z}_s$ are independent given the domain index. By contrast, the proposed latent causal model assumes a more general setting where $\mathbf{z}_c$ and $\mathbf{z}_s$ are dependent given the domain index, as depicted by Figure 2. We will further verify the advantages of the proposed latent causal model in experiments, compared with Sun et al. (2021).

## 4 IDENTIFIABILITY ANALYSIS OF THE PROPOSED LATENT CAUSAL MODEL

In this section, we provide identifiability analysis for the proposed latent causal model. We first build a connection between the proposed latent causal model and nonlinear ICA by using the independence among latent noise variables. We then show that it is still challenging to completely identify the proposed latent causal model due to *transitivity*, even with the identifiability result in nonlinear ICA. We finally show how to partially identify the latent content variable $\mathbf{z}_c$, by integrating the identifiability result of nonlinear ICA (Khemakhem et al., 2020) with the dependence between $\mathbf{n}_c$ and $\mathbf{y}$.

### 4.1 RELATING THE PROPOSED LATENT CAUSAL MODEL WITH NONLINER ICA

The proposed latent causal model splits latent noise variables $\mathbf{n}$ into two disjoint parts, $\mathbf{n}_c$ and $\mathbf{n}_s$, as depicted by Figure 3. Since $n_i$ models the noise information, the latent noise variables $n_i$ are assumed to be independent with each other in a causal system (Pearl, 2000; Spirtes et al., 2001)[1]. As a result, it is natural to connect the latent noise variables $n_i$ with latent indepen-

---

[1]For convenience in the later parts, with a slight abuse of definition, independent $n_i$ means that $n_i$ are mutually independent conditional on the observed variable $D$.

dent variables in nonlinear ICA. Specifically, nonlinear ICA aims to separate independent latent variables conditional on $\mathbf{D}$, *e.g.*, $n_i$, from observed mixing data, *e.g.*, $\mathbf{x}$, generated by a nonlinear function. Recent progress in Khemakhem et al. (2020) shows that one can recover $n_i$ up to permutation and scaling with relatively mild conditions, *e.g.*, there is an auxiliary observed variable, similar as $D$ in Figure 3, which modulates the distributions of all independent latent variables $n_i$.

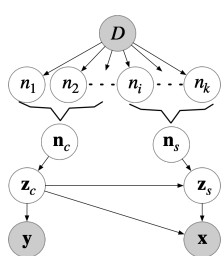

The identifiability result of $n_i$ holds in both source and target domains, since we have feature data $\mathbf{x}$ from both source and target domains in the training phase for domain adaptation. However, the permutation indeterminacy implies that we can not determine which recovered variables $n_i$ correspond to $\mathbf{n}_c$ (or $\mathbf{n}_s$) without further information. Therefore, using the identifiability result of nonlinear ICA only is insufficient to identify $\mathbf{n}_c$ and $\mathbf{n}_s$. We will further discuss how to handle the permutation indeterminacy in Section 4.3. Before that, we first analyze identifiability of the proposed latent causal model to provide a deep insight.

Figure 3: Relating with Nonliner ICA.

## 4.2 Complete Identifiability: The Non-identifiability Result

Even with the identifiability result of nonlinear ICA, it is still challenging to completely identify the proposed latent causal model. In particular, we have the following result:

**Proposition 4.1.** *With the identifiability result of nonlinear ICA under certain assumptions [2], the proposed latent causal model is still unidentifiable without additional assumptions, due to transitivity in latent space.*

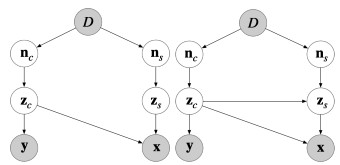

Figure 4: Two equivalent graph structures.

Intuitively, the non-identifiability result above is because we can not determine which path is the correct path corresponding to the net effect of $\mathbf{n}_c$ on $\mathbf{x}$; e.g., $\mathbf{n}_c \to \mathbf{z}_c \to \mathbf{x}$ and $\mathbf{n}_c \to \mathbf{z}_c \to \mathbf{z}_s \to \mathbf{x}$ are equivalent causal structures in Figure 4. This indeterminacy is due to *transitivity*, because if $\mathbf{z}_c \to \mathbf{z}_s \to \mathbf{x}$, then $\mathbf{z}_c \to \mathbf{x}$ can also be an alternative graph structure to generate same observed data, without further assumptions. For example, for simplicity, let us only consider one-dimensional $z_c$ and $z_s$. According to the graph structure shown in the right column of Figure 4, assume that $z_c := n_c$, $z_s := z_c + n_c$ and $\mathbf{x} := \mathbf{f}(z_1, z_2) + \boldsymbol{\varepsilon}$ (case 1). We then consider the graph structure shown in the left column of Figure 4, assume that $z_c := n_c$, $z_s := n_c$ and $\mathbf{x} := \mathbf{f} \circ \mathbf{g}(z_c, z_s) + \boldsymbol{\varepsilon}$ where $\mathbf{g}(z_c, z_s) = [z_c, z_c + z_s]$ (case 2). Interestingly, we find that the causal models in case 1 and case 2 generate the same observed data $\mathbf{x}$, which implies that there are two different causal models to interpret the same observed data, even with the identifiable $n_c$ and $n_s$. It often appears in latent causal discovery and seriously hinders the identifiability of latent causal models. For reader who may be interested in that problem, we recommend recent work by Adams et al. (2021).

## 4.3 Partial Identifiability: Identifying $\mathbf{z}_c$ up to Scaling

Although completely identifying the proposed latent causal model is challenging, for domain adaptation application, we are only interested in the identifiability of $\mathbf{z}_c$, instead of the latent style variable $\mathbf{z}_s$, since label $\mathbf{y}$ is only caused by $\mathbf{z}_c$. Thanks to the observed $\mathbf{y}^{\mathcal{S}}$ from source domains, we have the following identifiability result:

**Proposition 4.2.** *Assume that*

*(i) all latent noise variables $n_i$ can be identified up to permutation and scaling,*

---

[2]See APPENDIX A.3 for detailed assumptions.

*the latent content variable $\mathbf{z}_c$ in the proposed latent causal model can be identifiable up to scaling by using the dependence between $\mathbf{n}_c^{\mathcal{S}}$ and $\mathbf{y}^{\mathcal{S}}$ from source domains.*

Condition (i) can be obtained from the identifiability result of nonlinear ICA (Khemakhem et al., 2020). The proposition shows that the content variable $\mathbf{z}_c$ can be identified up to scaling by using the identifiability result of nonlinear ICA and the dependence between $\mathbf{y}$ and $\mathbf{n}_c$, in which the identifiability result of nonlinear ICA ensures to recover $n_i$ up to permutation and scaling, while the dependence removes the permutation indeterminacy to identify $\mathbf{n}_c$ and thus $\mathbf{z}_c$. The scaling indeterminacy of $\mathbf{z}_c$ results from the scaling indeterminacy of $n_i$ and the unknown mapping form $\mathbf{n}_c$ to $\mathbf{z}_c$. The scaling indeterminacy of $\mathbf{z}_c$ has no significance and can be ignored in latent space, since it can be 'absorbed' by the nonlinear mapping from $\mathbf{z}_c$ to $\mathbf{y}$. For example, consider the recovered variable $\hat{\mathbf{z}}_c$ and its scaling $scaling(\hat{\mathbf{z}}_c)$. When we try to learn a invariant predictor $g(\cdot)$ form $\hat{\mathbf{z}}_c$ to $\mathbf{y}$, the scaling indeterminacy can be 'absorbed' by a composition predictor, e.g., $g(scaling(\cdot))$.

## 5 LEARNING INVARIANT $p_D(\mathbf{y}|\mathbf{z}_c)$ FOR MSDA

The identifiable $\mathbf{z}_c$ provides a principled way to guarantee that we can learn the conditional distribution $p_D(\mathbf{y}|\mathbf{z}_c)$ that is invariant across domains and thus can be generalized to the target domain. Furthermore, since the identifiable $\mathbf{n}_c$ is the only parent node of $\mathbf{z}_c$, learning $p_D(\mathbf{y}|\mathbf{z}_c)$ can be transferred into learning $p_D(\mathbf{y}|\mathbf{n}_c)$, which is also invariant across domains as depicted by Figure 2. In this section, we propose a novel method to show how to learn the invariant conditional distribution $p_D(\mathbf{y}|\mathbf{n}_c)$ for MSDA.

### 5.1 THE PROPOSED METHOD FOR LEARNING INVARIANT $p_D(\mathbf{y}|\mathbf{n}_c)$

As analyzed in Section 4.3, the identifiability of $\mathbf{z}_c$ is based on the identifiability result of nonlinear ICA, so we need to identify $n_i$ first. To meet the conditions of identifiable $n_i$ as mentioned in Khemakhem et al. (2020), we employ the following Gaussian prior on $\mathbf{n}_c$ and $\mathbf{n}_s$:

$$p(\mathbf{n}|D) = p(\mathbf{n}_c|D)p(\mathbf{n}_s|D) = \mathcal{N}\big(\boldsymbol{\mu}_{\mathbf{n}_c}(D), \Sigma_{\mathbf{n}_c}(D)\big)\mathcal{N}\big(\boldsymbol{\mu}_{\mathbf{n}_s}(D), \Sigma_{\mathbf{n}_s}(D)\big), \qquad (1)$$

where $\boldsymbol{\mu}$ and $\Sigma$ denote the mean and variance, respectively. Both depend on the domain variable $D$ and can be implemented with multi-layer perceptrons. Since $n_i$ are to independent noise variables, $\Sigma$ here is a diagonal matrix. Some other exponential distributions, *e.g.*, Laplace distribution, also meet the conditions of identifiable $n_i$ and thus are also feasible (Khemakhem et al., 2020). We here use the Gassian prior since it is easy to leverage the reparameterization trick (Kingma & Welling, 2013). The proposed Gaussian prior equation 1 gives rise to the following variational posterior:

$$q(\mathbf{n}|D, \mathbf{x}) = q(\mathbf{n}_c|D, \mathbf{x})q(\mathbf{n}_s|D, \mathbf{x}) = \mathcal{N}\big(\boldsymbol{\mu}'_{\mathbf{n}_c}(\mathbf{D}, \mathbf{x}), \Sigma'_{\mathbf{n}_c}(\mathbf{D}, \mathbf{x})\big)\mathcal{N}\big(\boldsymbol{\mu}'_{\mathbf{n}_s}(\mathbf{D}, \mathbf{x}), \Sigma'_{\mathbf{n}_s}(\mathbf{D}, \mathbf{x})\big),$$
$$(2)$$

where $\boldsymbol{\mu}'$ and $\Sigma'$ denote the mean and variance of the posterior, respectively; both of them depend on the domain variable $D$ and the observed $\mathbf{x}$ and can be implemented by multi-layer perceptrons. Combining this with the Gaussian prior in equation 1, we can derive the following evidence lower bound (ELBO):

$$\max \mathbb{E}_{q(\mathbf{n}|D, \mathbf{x})}\big(p(\mathbf{x}|D)\big) - D_{\mathcal{KL}}\big(q(\mathbf{n}|D, \mathbf{x})||p(\mathbf{n}|D)\big), \qquad (3)$$

where $D_{\mathcal{KL}}$ denotes the Kullback–Leibler divergence.

By maximizing the ELBO equation 3, we can then recover $n_i$ up to scaling and permutation. To remove the permutation indeterminacy as mentioned in Section 4.3, we can simultaneously maximize the correlation between $\mathbf{y}^{\mathcal{S}}$ and $n_i^{\mathcal{S}}$ to identify $\mathbf{n}_c^{\mathcal{S}}$. Here we employ the mutual information to maximize the dependence. As a result, we arrive at:

$$\max \lambda \Big( \underbrace{\mathbb{E}_{q(\mathbf{n}|D, \mathbf{x})}\big(p(\mathbf{x}|D)\big) - D_{\mathcal{KL}}\big(q(\mathbf{n}|D, \mathbf{x})||p(\mathbf{n}|D)\big)}_{ELBO} \Big) + \underbrace{I(\mathbf{n}_c^{\mathcal{S}}, \mathbf{y}^{\mathcal{S}})}_{MI}, \qquad (4)$$

where $I(\mathbf{n}_c^{\mathcal{S}}, \mathbf{y}^{\mathcal{S}})$ denotes the mutual information between $\mathbf{n}_c^{\mathcal{S}}$ and $\mathbf{y}^{\mathcal{S}}$ in source domains, and $\lambda$ is a regularization hyper-parameter that balances the ELBO and the mutual information (MI). The proposed method is termed iLCC-MSDA (identifiable Latent Causal Content for MSDA), including two components: ELBO and mutual information. The ELBO component ensures that $n_i$ can be recovered up to scaling and permutation. The MI component handles the permutation, and thus ensures

which recovered $n_i$ corresponds to the latent content variables $\mathbf{n}_c$. In the implementation, we use the variational low bounder of mutual information proposed by Alemi et al. (2016) to approximate the mutual information in equation 4.

$$\max \lambda \Big( \mathbb{E}_{q(\mathbf{n}|D,\mathbf{x})}\big(p(\mathbf{x}|D)\big) - D_{\mathcal{KL}}\big(q(\mathbf{n}|D,\mathbf{x})||p(\mathbf{n}|D)\big) \Big) + \mathbb{E}_{q(\mathbf{n}_c|D,\mathbf{x})}\big(p(\mathbf{y}^{\mathcal{S}}|\mathbf{n}_c^{\mathcal{S}})\big). \quad (5)$$

A graphical depiction of the proposed iLCC-MSDA is shown in Figure 5.

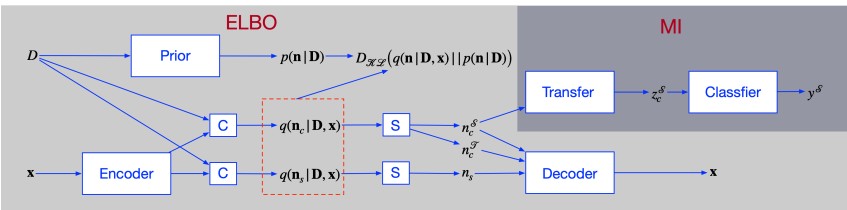

Figure 5: The proposed iLCC-MSDA to learn the invariant $p(\mathbf{y}|\mathbf{n}_c)$ for multiple source domain adaptation. C denotes concatenation, and S denotes sampling from the posterior distributions.

**Constraining the independence among $n_i$** As we discussed, the performance of the proposed iLCC-MSDA above relies on assumptions on the identifiability of $n_i$, which requires that there are enough domains across which $n_i$ changes in order to well capture the statistical independence. However, in real applications, we may not have sufficient domains. To mitigate this issue, motivated by disentangled representations (Higgins et al., 2017; Kim & Mnih, 2018; Chen et al., 2018), we proposes to use a hyperparameter $\beta$ to enhance the independence among $n_i$.

**Entropy regularization** In the loss function in equation 5, we maximize the causal influence between $\mathbf{y}^{\mathcal{S}}$ and $\mathbf{n}_c^{\mathcal{S}}$ in source domains with mutual information. To encourage such causal influence in target domain, we can also maximize the mutual information between $\hat{\mathbf{y}}^{\mathcal{T}}$ and $\mathbf{n}_c^{\mathcal{T}}$ by minimizing the following conditional entropy:

$$L_{ent} = -\mathbb{E}\Big( p(\hat{\mathbf{y}}^{\mathcal{T}}|\mathbf{n}_c^{\mathcal{T}}) \log p(\hat{\mathbf{y}}^{\mathcal{T}}|\mathbf{n}_c^{\mathcal{T}}) \Big), \quad (6)$$

where $\hat{\mathbf{y}}^{\mathcal{T}}$ denotes the estimated label in the target domain. This regularization has been empirically used to make label predictions more deterministic in previous works (Wang et al., 2020; Li et al., 2021), while we consider it from the view of causality.

Therefore, our final loss function is:

$$\max \lambda(\mathbb{E}_{q(\mathbf{n}|D,\mathbf{x})}(p(\mathbf{x}|D)) - \beta D_{\mathcal{KL}}\big(q(\mathbf{n}|D,\mathbf{x})||p(\mathbf{n}|D)\big)) + \mathbb{E}_{q(\mathbf{n}_c|D,\mathbf{x})}\big(p(\mathbf{y}^{\mathcal{S}}|\mathbf{n}_c^{\mathcal{S}})\big) + \gamma L_{ent}, \quad (7)$$

where $\beta, \lambda, \gamma$ are hyper-parameters that trade off the independence of $\mathbf{n}_c$ and $\mathbf{n}_s$, the classifier and the entropy regularization loss terms.

## 6 EXPERIMENTS

### 6.1 EXPERIMENTS ON SYNTHETIC DATA

**Dataset** We conduct experiments on synthetic data, generated by the following process: we divide the latent variables into 5 segments, which are corresponding to 5 domains. Each segment includes 1000 examples. Within each segment, we first sample the mean and the variance from uniform distributions $[1, 2]$ and $[0.3, 1]$ for the latent exogenous variables $\mathbf{n}_c$ and $\mathbf{n}_s$, respectively. Then for each segment, we generate $\mathbf{z}_c, \mathbf{z}_s, \mathbf{x}$ and $\mathbf{y}$ according to the following structural causal model:

$$\mathbf{z}_c := \mathbf{n}_c, \qquad \mathbf{z}_s := \mathbf{z}_c^3 + \mathbf{n}_s, \qquad \mathbf{y} := \mathbf{z}_c^3, \qquad \mathbf{x} := MLP(\mathbf{z}_c, \mathbf{z}_s), \quad (8)$$

where following (Khemakhem et al., 2020) we mix the latent $\mathbf{z}_c$ and $\mathbf{z}_s$ using a multi-layer perceptron to generate $\mathbf{x}$.

**Results** In implementation, we use the first 4 segments as source domains, and the last segment as target domain. Figure 6(a) shows the true and recovered distributions of the exogenous variables $\mathbf{n}_c$. Due to the support of nonlinear ICA, the proposed iLCC-MSDA obtain the mean correlation coefficient (MCC) 0.96 between the original $\mathbf{n}_c$ and the recovered. Due to the invariant conditional distribution $p(\mathbf{y}|\mathbf{n}_c)$, even with the change of distribution of the exogenous variables $\mathbf{n}_c$ as shown in Figure 6(a), the learned $p(\mathbf{y}|\mathbf{n}_c)$ can generalize to target segment in a principle way as depicted by the Figure 6(b). Due to the limited space, Figure 6(b) only shows 200 samples for the true and predicted $\mathbf{y}$.

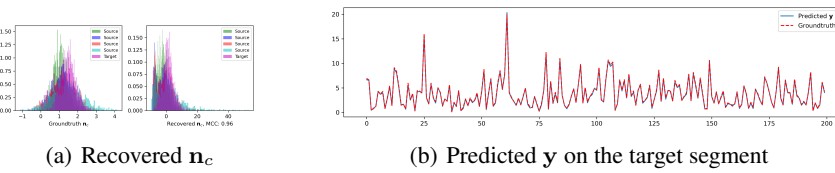

(a) Recovered $\mathbf{n}_c$         (b) Predicted $\mathbf{y}$ on the target segment

Figure 6: The Result on Synthetic Data.

### 6.2 EXPERIMENTS ON REAL DATA

**Dataset** We further evaluate the proposed iLCC-MSDA on benchmark domain adaptation dataset PACS dataset (Li et al., 2017) and Terra Incognita (Beery et al., 2018). In the original PACS, the label distributions for any two domains is very similar (*i.e.*, $D_{\mathcal{KL}} \approx 0.1$). This data is suitable for domain adaptation with conditional shift as shown in Figure 1 (b) where the label distribution remains unchanged, while it is not appropriate for the proposed setting where the label distribution changes across domains, as shown in Figure 1 (c). Therefore, we randomly sample the original PACS dataset to provide new PACS dataset where the label distribution changes, to generate three datasets, PACS ($D_{\mathcal{KL}} = 0.3$), PACS ($D_{\mathcal{KL}} = 0.5$) and PACS ($D_{\mathcal{KL}} = 0.7$). Here $D_{\mathcal{KL}} = 0.3(0.5, 0.7)$ denotes that KL divergence of label distributions in any two different domains is approximately 0.3 (0.5, 0.7). See APPENDIX for details of label distributions.

**Baselines** We compare the proposed method with state-of-the-art methods to verify its effectiveness. Particularly, we compare the proposed methods with empirical risk minimization (ERM), MCDA (Saito et al., 2018), M3DA (Peng et al., 2019), LtC-MSDA (Wang et al., 2020), T-SVDNet (Li et al., 2021), IRM (Arjovsky et al., 2019), IWCDAN (Tachet des Combes et al., 2020) and LaCIM (Sun et al., 2021). In these methods, MCDA, M3DA, LtC-MSDA and T-SVDNet learn invariant representations for MSDA, while IRM, IWCDAN and LaCIM learn invariant conditional distributions, allowing label distribution to change. Details of implementation, including network architectures and hyper-parameter setting, are in the APPENDIX. All the proposed methods are averaged over 3 runs with standard deviation.

**Ablation studies** The bottom of Table 1 and 2 presents the results for ablation studies. We can observe that entropy regularization equation 6 significantly increases the performance (around 10% and 5%) of the proposed method on both dataset. This justifies the importance of the causal relation between $\mathbf{y}$ and $\mathbf{n}_c$, which is consistent with our model assumption. Besides, the hyper-parameter $\beta$ also boosts the performance by enforcing the independence among the latent variables $n_i$.

**Results** The results by different methods on PACS are presented in Table 1. We can observe that as the increase of KL divergence of label distribution, the performance of MCDA, M3DA, LtC-MSDA and T-SVDNet, which are based on learning an invariant representations, gradually degenerates. When the KL divergence is about 0.7, the performance of these methods is worse than traditional ERM. Compared with IRM, IWCDAN and LaCIM, which allows label distribution to change across domains, the proposed iLCC-MSDA obtains the best performance, due to our theoretical supports. Table 2 depicts the results by different methods on challenging Terra Incognit. The proposed iLCC-MSDA achieves a significant performance gain on the challenging task $\rightarrow$L7. Compared with the other methods, the proposed iLCC-MSDA is the only one that is superior to ERM.

## 7 CONCLUSION

The key for domain adaptation is to understand how the joint distribution of features and label changes across domains. Previous works usually assume covariate shift or conditional shift to inter-

Table 1: Classification results and ablation study on PACS data.

| PACS ($D_{\mathcal{KL}} = 0.3$) | | | | | |
|---|---|---|---|---|---|
| **Methods** | Accuracy | | | | |
| | →Art | →Cartoon | →Photo | →Sketch | Average |
| ERM | $82.3 \pm 0.3$ | $81.3 \pm 0.9$ | $94.9 \pm 0.2$ | $76.2 \pm 0.7$ | 83.6 |
| MCDA ((Saito et al., 2018)) | $76.6 \pm 0.6$ | $85.1 \pm 0.3$ | $96.6 \pm 0.1$ | $70.1 \pm 1.3$ | 82.1 |
| M3SDA (Peng et al., 2019) | $79.6 \pm 1.0$ | $\mathbf{86.6 \pm 0.5}$ | $\mathbf{97.1 \pm 0.3}$ | $83.3 \pm 1.0$ | 86.6 |
| LtC-MSDA (Wang et al., 2020) | $82.7 \pm 1.3$ | $84.9 \pm 1.4$ | $96.9 \pm 0.2$ | $75.3 \pm 3.1$ | 84.9 |
| T-SVDNet (Li et al., 2021) | $81.8 \pm 0.3$ | $86.5 \pm 0.2$ | $95.9 \pm 0.2$ | $80.7 \pm 0.8$ | 86.3 |
| IRM (Arjovsky et al., 2019) | $79.6 \pm 0.7$ | $77.0 \pm 2.2$ | $94.6 \pm 0.2$ | $71.7 \pm 2.3$ | 80.7 |
| IWCDAN (Tachet des Combes et al., 2020) | $84.0 \pm 0.5$ | $78.1 \pm 0.7$ | $96.0 \pm 0.1$ | $75.5 \pm 1.9$ | 83.4 |
| LaCIM (Sun et al., 2021) | $63.1 \pm 1.5$ | $72.6 \pm 1.0$ | $82.7 \pm 1.3$ | $71.5 \pm 0.9$ | 72.5 |
| iLCC-MSDA(Ours) | $\mathbf{86.4 \pm 0.8}$ | $81.1 \pm 0.8$ | $95.9 \pm 0.1$ | $\mathbf{86.0 \pm 1.0}$ | **87.4** |
| PACS ($D_{\mathcal{KL}} = 0.5$) | | | | | |
| ERM | $85.4 \pm 0.6$ | $76.4 \pm 0.5$ | $94.4 \pm 0.4$ | $85.0 \pm 0.6$ | 85.3 |
| MCDA ((Saito et al., 2018)) | $81.6 \pm 0.1$ | $76.8 \pm 0.1$ | $93.6 \pm 0.1$ | $84.1 \pm .6$ | 84.0 |
| M3SDA (Peng et al., 2019) | $81.2 \pm 1.2$ | $77.5 \pm 1.3$ | $94.5 \pm 0.5$ | $84.3 \pm 0.5$ | 84.4 |
| LtC-MSDA (Wang et al., 2020) | $85.2 \pm 1.5$ | $75.2 \pm 2.6$ | $94.9 \pm 0.6$ | $85.1 \pm 2.7$ | 85.1 |
| T-SVDNet (Li et al., 2021) | $84.8 \pm 0.3$ | $77.6 \pm 1.7$ | $94.2 \pm 0.2$ | $86.4 \pm 0.2$ | 85.6 |
| IRM (Arjovsky et al., 2019) | $81.5 \pm 0.3$ | $71.1 \pm 1.3$ | $94.2 \pm 0.1$ | $78.7 \pm 0.7$ | 81.4 |
| IWCDAN (Tachet des Combes et al., 2020) | $79.2 \pm 1.6$ | $72.6 \pm 0.7$ | $\mathbf{95.6 \pm 0.1}$ | $82.1 \pm 2.2$ | 82.4 |
| LaCIM (Sun et al., 2021) | $67.4 \pm 1.6$ | $66.6 \pm 0.6$ | $81.0 \pm 1.2$ | $82.3 \pm 0.6$ | 74.3 |
| iLCC-MSDA(Ours) | $\mathbf{89.0 \pm 0.7}$ | $\mathbf{77.6 \pm 0.5}$ | $95.0 \pm 0.3$ | $\mathbf{87.4 \pm 1.6}$ | **87.3** |
| PACS ($D_{\mathcal{KL}} = 0.7$) | | | | | |
| ERM | $86.1 \pm 0.6$ | $\mathbf{76.8 \pm 0.3}$ | $94.6 \pm 0.4$ | $81.3 \pm 2.0$ | 84.7 |
| MCDA ((Saito et al., 2018)) | $80.8 \pm 0.7$ | $74.1 \pm 1.2$ | $94.4 \pm 0.4$ | $77.9 \pm 0.4$ | 81.8 |
| M3SDA (Peng et al., 2019) | $82.7 \pm 1.3$ | $76.2 \pm 1.0$ | $94.5 \pm 0.7$ | $80.8 \pm 1.2$ | 83.6 |
| LtC-MSDA (Wang et al., 2020) | $83.7 \pm 1.6$ | $74.6 \pm 1.4$ | $95.0 \pm 0.7$ | $80.8 \pm 0.6$ | 83.5 |
| T-SVDNet (Li et al., 2021) | $83.3 \pm 0.8$ | $74.7 \pm 0.6$ | $95.2 \pm 0.3$ | $74.5 \pm 3.3$ | 81.9 |
| IRM (Arjovsky et al., 2019) | $84.3 \pm 0.8$ | $73.3 \pm 1.8$ | $94.3 \pm 0.1$ | $69.4 \pm 4.6$ | 80.3 |
| IWCDAN (Tachet des Combes et al., 2020) | $76.3 \pm 0.8$ | $73.9 \pm 1.6$ | $93.1 \pm 0.5$ | $77.6 \pm 3.8$ | 80.2 |
| LaCIM (Sun et al., 2021) | $63.6 \pm 0.9$ | $68.7 \pm 1.4$ | $77.5 \pm 3.8$ | $77.8 \pm 2.2$ | 71.9 |
| iLCC-MSDA(Ours) | $\mathbf{90.7 \pm 0.3}$ | $74.2 \pm 0.7$ | $\mathbf{95.8 \pm 0.3}$ | $\mathbf{83.0 \pm 2.2}$ | **86.0** |
| iLCC-MSDA(Ours) with $\beta = 1$ | $90.2 \pm 0.5$ | $73.4 \pm 0.8$ | $95.7 \pm 0.4$ | $82.7 \pm 0.7$ | 85.5 |
| iLCC-MSDA(Ours) with $\gamma = 0$ | $81.1 \pm 1.5$ | $70.0 \pm 1.6$ | $92.0 \pm 0.5$ | $59.6 \pm 0.7$ | 75.7 |

Table 2: Classification results on TerraIncognita.

| **Methods** | Accuracy | | | | |
|---|---|---|---|---|---|
| | →L28 | →L43 | →L46 | →L7 | Average |
| ERM | $54.1 \pm 2.8$ | $62.3 \pm 0.7$ | $44.7 \pm 0.9$ | $74.5 \pm 2.6$ | 58.9 |
| MCDA ((Saito et al., 2018)) | $54.9 \pm 4.1$ | $61.2 \pm 1.2$ | $42.7 \pm 0.3$ | $64.8 \pm 8.1$ | 55.9 |
| M3SDA (Peng et al., 2019) | $62.3 \pm 1.4$ | $62.7 \pm 0.4$ | $41.3 \pm 0.3$ | $57.4 \pm 0.9$ | 55.9 |
| LtC-MSDA (Wang et al., 2020) | $51.9 \pm 5.7$ | $54.6 \pm 1.3$ | $45.7 \pm 1.0$ | $69.1 \pm 0.3$ | 55.3 |
| T-SVDNet (Li et al., 2021) | $58.2 \pm 1.7$ | $61.9 \pm 0.3$ | $45.6 \pm 2.0$ | $68.2 \pm 1.1$ | 58.5 |
| IRM (Arjovsky et al., 2019) | $57.5 \pm 1.7$ | $60.7 \pm 0.3$ | $42.4 \pm 0.6$ | $74.1 \pm 1.6$ | 58.7 |
| IWCDAN (Tachet des Combes et al., 2020) | $58.1 \pm 1.8$ | $59.3 \pm 1.9$ | $43.8 \pm 1.5$ | $58.9 \pm 3.8$ | 55.0 |
| LaCIM (Sun et al., 2021) | $58.2 \pm 3.3$ | $59.8 \pm 1.6$ | $46.3 \pm 1.1$ | $70.8 \pm 1.0$ | 58.8 |
| iLCC-MSDA(Ours) | $\mathbf{64.3 \pm 3.4}$ | $\mathbf{63.1 \pm 1.6}$ | $44.7 \pm 0.4$ | $\mathbf{80.8 \pm 0.4}$ | **63.2** |
| iLCC-MSDA(Ours) with $\beta = 1$ | $56.3 \pm 4.3$ | $61.5 \pm 0.7$ | $45.2 \pm 0.3$ | $80.1 \pm 0.6$ | 60.8 |
| iLCC-MSDA(Ours) with $\gamma = 0$ | $54.8 \pm 1.4$ | $58.9 \pm 1.8$ | $\mathbf{46.8 \pm 1.4}$ | $73.1 \pm 0.6$ | 58.4 |

pret the change of the joint distribution, which may be restricted in some real applications. Hence, this work considers a new and milder assumption, latent covariate shift. Specifically, we propose a latent causal model to precisely formulate the generative process of input features and label. We show that the latent content variable in the proposed latent causal model can be identified up to scaling. This inspires a new method to learn the invariant label distribution conditional on the latent causal variable, resulting in a principled way to guarantee generalization to target domains. Experiments demonstrate the theoretical results and the efficacy of the proposed method, compared with state-of-the-art methods across various data sets.

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

## A   APPENDIX

**Data Details**   The commonly used datasets for multi-source domain adaptation, such as Digits-five, Office-Home, PACS, and DomainNet, are not considered in this work, because for these dataset the label distributions of any two domains is very similar, which is suitable for domain adaptation with conditional shift as shown in Figure 1 (b). However, these datasets are not appropriate for the proposed setting where the label distribution changes across domains, as shown in Figure (c). Therefore, we resample the original PACS (Li et al., 2017) dataset, which contains 4 domains, Photo, Artpainting, Cartoon and Sketch, which shares the same seven categories. The KL divergence of label distributions of any two domains in the original PACS is very small, round 0.1. For obtaining dataset that meets the requirement of the proposed latent covariate shift, we filter the original dataset by re-sampling it, and obtain three new datasets with different the KL divergences of label distributions as depicted by Figure 7. The resampling process just randomly select some sample from the original PACS dataset, so that the labels distribution changes across domains. The labels distribution are depicted by Figure 7 . For Terra Incognita (Beery et al., 2018), the label distribution is long-tailed at each domain, and each domain has a different label distribution, which is naturally applicable for our setting. This work uses the four domains from the original data, L28, L43, L46 and L7, which shares the same seven categories: bird, bobcat, empty, opossum, rabbit, raccoon, skunk, as depicted by Figure 8. (Here 'L8' denotes the image data is collected from the location 28.)

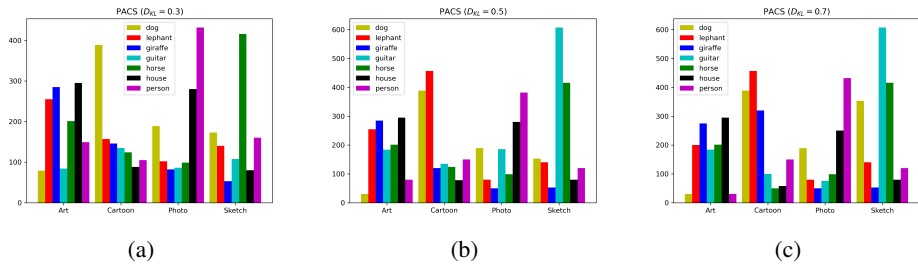

(a)               (b)               (c)

Figure 7: Label distributions of the filtered PACS data.

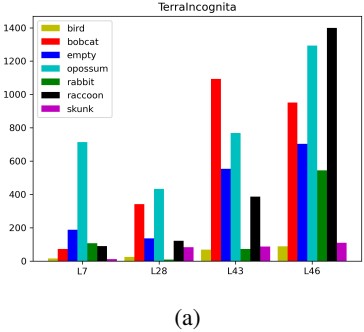

(a)

Figure 8: Label distributions of Terra Incognita data used in this work.

**Implementation Details**   For the synthetic data, we used a encoder, *e.g.* 3-layer fully connected network with 30 hidden nodes for each layer, and decoder, *e.g.* 3-layer fully connected network with 30 hidden nodes for each layer. We use 3-layer fully connected network with 30 hidden nodes for prior model. Since this is a ideal environment to verify the proposed method, for hyper-parameters, we set $\beta = 1$ and $\gamma = 0$ to remove the heuristic constraints, and we set $\lambda = 1e-2$. For the real data, all methods used the same network backbone, ResNet-18 pre-trained on ImageNet. Since it can be challenging to train VAE on high-resolution images, we use extracted features by ResNet-18 as our VAE input. We then use 2-layer fully connected networks as the VAE encoder and decoder, use

2-layer fully connected network for the prior model, use 2-layer fully connected network to transfer $\mathbf{n}_c$ to $\mathbf{z}_c$. For hyper-parameters, we set $\beta = 4$, $\gamma = 0.1$, $\lambda = 1e - 4$ for the proposed method on all datasets.

## A.1 THE PROOF OF PROPOSITION 4.1

To prove non-identifiability, it is sufficient to show that several different graph structures can lead to the same observed data. In particular, let us consider the net effect of $\mathbf{n}_c$ on $\mathbf{x}$. There are two different paths to 'explain' the net effect of $\mathbf{n}_c$ on $\mathbf{x}$. One path is $\mathbf{n}_c \rightarrow \mathbf{z}_c \rightarrow \mathbf{x}$. In this case, since we have no limitation on the function class of edges, we can cut the path $\mathbf{z}_c \rightarrow \mathbf{z}_s$ off (e.g., the left sub-figure of Figure 4) and obtain the same observed data depicted by the right sub-figure of Figure 4. Therefore, there is a causal graph depicted by the left sub-figure of Figure 4 are equivalent with the proposed latent causal model as depicted by the right sub-figure of Figure 4.

## A.2 THE PROOF OF PROPOSITION 4.2

As mentioned in Section 4.1, there are the permutation indeterminacy and scaling indeterminacy in identifiable $n_i$. The permutation indeterminacy implies that we are uncertain of which the recovered variable $\hat{n}_i$ are corresponding to the latent content variable $\mathbf{n}_c$. If we can solve this permutation problem, since the parent node of $\mathbf{z}_c$ includes $\mathbf{n}_c$ only, $\mathbf{z}_c$ can be identifiable up to scaling, i.e., $\mathbf{z}_c = f(\mathbf{n}_c)$ where $f$ can be any nonlinear function. Let us consider the relationships between each $\hat{n}_i$ and $\mathbf{y}$ in the proposed causal model, it is clear that the label $\mathbf{y}$ depends on $\mathbf{n}_c$, and is independent with $\mathbf{n}_s$, given the domain variable $D$. As a result, we can compute the correlations (*e.g.*, by mutual information) between $\mathbf{y}^{\mathcal{S}}$ from source domains and $\hat{n}_i^{\mathcal{S}}$ from source domains to determine which recovered variables $\hat{n}_i$ are corresponding to the latent content variable $\mathbf{n}_c$.

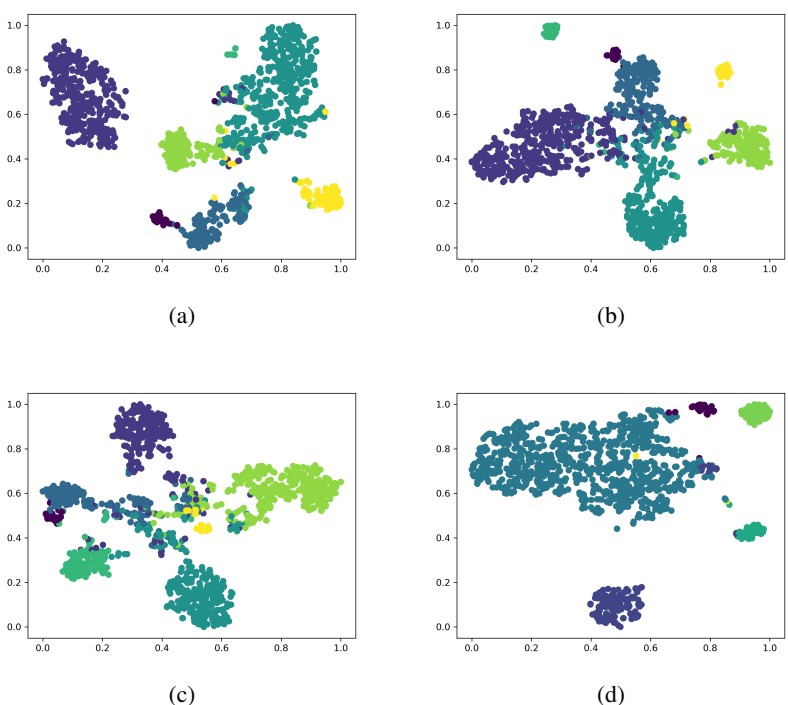

Figure 9: The t-SNE visualizations of learned features $\mathbf{n}_c$ of different domains on the $\rightarrow$L7 task in TerraIncognita. (a) The learned features $\mathbf{n}_c$ in the L28 domain (b) The learned features $\mathbf{n}_c$ in the L43 domain (c) The learned features $\mathbf{n}_c$ in the L46 domain (d)The learned features $\mathbf{n}_c$ in the L7 domain. We can observe that the distribution of learned feature $\mathbf{n}_c$ by the proposed method changes across domains, which is very different with the previous methods based on learning invariant representations.

### A.3 ASSUMPTIONS FOR NONLINEAR ICA

Our theoretical results, *e.g.*, Propositions 4.1 and 4.2, rely on the identifiability result of nonlinear ICA. For completeness, we present assumptions for the identifiability result Khemakhem et al. (2020); Sorrenson et al. (2020), as follows:

**Theorem A.1.** *Khemakhem et al. (2020); Sorrenson et al. (2020) Suppose the following conditional generative model:*

$$n_i :\sim \mathcal{N}(\beta_{i,1}(D), \beta_{i,2}(D)), \tag{9}$$
$$\mathbf{x} := \mathbf{f}(\mathbf{n}) + \boldsymbol{\varepsilon}. \tag{10}$$

*Assume the following holds:*

(i) *The set $\{\mathbf{x} \in \mathcal{X} | \varphi_\varepsilon(\mathbf{x}) = 0\}$ has measure zero (i.e., has at most countable number of elements), where $\varphi_\varepsilon$ is the characteristic function of the density $p_\varepsilon$.*

(ii) *The function $\mathbf{f}$ in Eq. 10 is bijective.*

(iii) *There exist $2d + 1$ distinct points $D_0, d_1, ..., D_{2d}$ such that the matrix*

$$\mathbf{L} = (\boldsymbol{\eta}(D_1) - \boldsymbol{\eta}(D_0), ..., \boldsymbol{\eta}(D_{2d}) - \boldsymbol{\eta}(D_0)) \tag{11}$$

*of size $2d \times 2d$ is invertible, where $d$ is the dimension of $\mathbf{n}$, $\boldsymbol{\eta}(D)$ denotes the vector of their coefficients, which depends on $\beta_{i,1}$ and $\beta_{i,2}$.*

*then the true latent variables $\mathbf{n}$ are related to the estimated latent variables $\hat{\mathbf{n}}$ by the following relationship:*

$$\mathbf{n} = \mathbf{P}\hat{\mathbf{n}} + \mathbf{c},$$

*where $\mathbf{P}$ denotes the permutation matrix with scaling, $\mathbf{c}$ denotes a constant vector.*

Eq. 9 is enforcing Gaussian distributions on the latent noise variables $\mathbf{n}$. Note that the assumptions of nonlinear ICA (Khemakhem et al., 2020) on the noise could be broad exponential family distribution, *e.g.*, Gaussian distributions, Laplace, Gamma distribution and so on. This work consider Gaussian distribution, mainly because we implement it in our experiment, as shown in Eqs. 1 and 2.

