# OpenReview forum: "Identifying Latent Causal Content for Multi-Source Domain Adaptation"
_ICLR.cc/2023/Conference — Submitted to ICLR 2023_

### Official Review · Reviewer_x2cf · 2022-10-21

**Confidence:** 3
**Correctness:** 3
**Technical Novelty And Significance:** 3
**Empirical Novelty And Significance:** 2
**Recommendation:** 5

**Clarity, Quality, Novelty And Reproducibility:**

The idea to solve the multi-source domain adaptation through non-linear identification is interesting, however, I think the current paper demonstration is not enough to support its claim.

**Strength And Weaknesses:**

This paper proposes to solve an important question in multi-source domain adaptation. Even though the latent content variable can only be identified together with the nonlinear ICA, the proposed framework shows the potential to solve this challenge. However, I think the current results section is weak, and the interpretation is not enough to demonstrate the superiority of this method.

**Summary Of The Paper:**

The authors argue that existing works on domain adaptation are restricted to assuming covariate shift or conditional shift to interpret the change of the joint distribution and thus have their limitations in real applications. They propose a new latent casual model to formulate the generative process of input features and labels, they evaluated their proposed approach on synthetic and two real datasets.

**Summary Of The Review:**

My overall impression is that the current experiments and results section is somewhat weak in supporting the paper's claim. For example:

1. The authors probably should consider adding evaluation metrics in the experiment section and why it is valid.

2. The introduction for the two real datasets is not clear to me, e.g. the motivation to choose these datasets; what does L7, L28, L43, and L46 mean in Terra Incognita?; because the authors modified the original PACS dataset to perform the experiments will that introduce inductive bias for the evaluation?

3. The Interpretation of the results is not convincing to me.  In Table 1, the authors argued that with the increase of KL divergence of label distribution, the performance of MCDA, M3DA(typo here, it should be M3SDA), LtC-MSDA and T-SVDNet gradually degenerates. However, in Table 1 we can see that MCDA and LtC-MSDA performed better when DKL=0.5 compared to DL=0.3 and then decreased under DKL=0.7 which is contradictory with the conclusions in the results. Secondly, there is only one sentence: "due to our theoretical supports", to support why iLCC-MSDA performs the best compared with IRM, IWCDAN and LaCIM.

4. L7 appeared in the results section very suddenly, it was not mentioned earlier and I feel it is very difficult to follow the logic there.

Besides, I also want to ask the authors how do you distinguish nc and ns, zc and zs in your model framwork?

---

> ### Author Response · Authors · 2022-11-18
> **Rebuttal**
>
> __*Q1:The authors probably should consider adding evaluation metrics in the experiment section and why it is valid.*__
>  It is common to only use accuracy for domain adaptation, e.g., Ref. MCDA ((Saito et al., 2018)), M3SDA (Peng et al., 2019),LtC-MSDA (Wang et al., 2020),T-SVDNet (Li et al., 2021).
>
> __*Q2:The introduction for the two real datasets is not clear to me, e.g. the motivation to choose these datasets; what does L7, L28, L43, and L46 mean in Terra Incognita?; because the authors modified the original PACS dataset to perform the experiments will that introduce inductive bias for the evaluation*__
> The commonly used datasets for multi-source domain adaptation, such as Digits-five, Office-Home, and DomainNet, are not considered in this work, because for these dataset the label distributions of any two domains are very similar. Therefore, these datasets are suitable for domain adaptation with conditional shift as shown in Figure 1 (b). However, these datasets are not appropriate for the proposed setting where the label distribution changes across domains, as shown in Figure (c). Therefore, we resample the original PACS dataset, and thus obtain new PACS datasets. The resampling process just randomly select some samples from the original PACS dataset, so that the labels distribution changes across domains. The resampling process do not introduce inductive bias. Terra Incognita datasat has been widely used in domain generalization, which is collected from different locations (e.g.,). L7, L28, L43, and L46 denotes the images from different locations. We have further clarifying these details, please see Data Details in APPENDIX.
>
> __*Q3:The Interpretation of the results is not convincing to me.*__
> Generally speaking, larger KL means that domain adaptation on this case is more changeling. However, due to various uncontrollable factors, e.g., the different sample sizes and local optimization, it is not necessarily means that we must have better results on larger KL case than results on smaller KL case, especially in the case where the change of KL is small (e.g., from 0.3 to 0.7). That is, we think the slightly fluctuations of performance are reasonable. Overall, our experimental results verify the proposed method is better than invariant representations based methods, e.g., MCDA, M3DA, LtC-MSDA and T-SVDNet, and the methods, including IRM, IWCDAN and LaCIM, which allows label distribution to change across domains. Such conclusion can be further verified from the results on dataset TerraIncognita.
>
> __*Q4: There is only one sentence: "due to our theoretical supports", to support why iLCC-MSDA performs the best compared with IRM, IWCDAN and LaCIM.*__
> Our work provides the identifiability result, providing a principled way to guarantee that the learned predictor can generalize to the target domain. In contrast, IRM and IWCDAN do not provide such guarantee. Although LaCIM provides similar identifiability result, we have clarified the disadvantages of LaCIM, as mentioned in the paragraph "*$\mathbf{z}_c$ *causes* $\mathbf{z}_s$*" in section 3. Please see that paragraph for details.
>
> __*Q5:L7 appeared in the results section very suddenly, it was not mentioned earlier and I feel it is very difficult to follow the logic there.*__
> In the original version, L7 appeared in the Table 2: Classification results on TerraIncognita, and also appeared in the data details in APPENDIX. We have further clarified these data details, please see Data Details in APPENDIX.
>
> __*Q6:Besides, I also want to ask the authors how do you distinguish nc and ns, zc and zs in your model framwork?*__
> 1. On model: For a causal system, the latent variables $\mathbf{n}$ represent some unmeasured factors that influence the latent causal variables $\mathbf{z}$. The latent noise variables $\mathbf{n}$ are assumed to be independent, conditional on the observed variable $D$, while there are causal relations among the latent causal variables $\mathbf{z}$, as shown in Figure 2.
> 2. On identifiability: The identifiability result of nonlinear ICA ensures recovering all $n_i$ up to permutation and scaling. From the graph structure in Figure 2, we can see that $y$ depends on $n_c$ only, so that we can measure this dependence (e.g., by mutual information) to identify which $n_i$ are corresponding to $n_c$. As a result, we can identify $n_c$ and $n_s$. Also, for the graph structure, $z_c$ is only caused by $n_c$, so that we can identifying $z_c$ up to a (nonlinear) transformation. Here the transformation depends on the assumption of the mapping from $n_c$ to $z_c$. Please see Proposition 4.2 for more details. We can not identify $z_s$, due to the transitivity. Please see Proposition 4.1 for more details.

---

> > ### Author Response · Authors · 2022-12-04
> > **A kind request for further feedback for your concern**
> >
> > Dear Reviewer x2cf,
> >
> > Your valuable comments have helped improve our presentation a lot. Thanks for providing the initial comments. We have provided responses to that. If there is any other concern, please let us know, and we will immediately respond to it. Your feedback is valuable to us. Thank you.

---

> > > ### Comment · Reviewer_x2cf · 2022-12-04
> > > **Response to the authors**
> > >
> > > Thank you for all the explanations. My concerns are mostly addressed, however, I also agree with reviewers fg87 and wpga's comments that the technical contribution is limited. So, I will keep my rate.

---

### Official Review · Reviewer_wpga · 2022-10-23

**Confidence:** 4
**Correctness:** 3
**Technical Novelty And Significance:** 2
**Empirical Novelty And Significance:** 2
**Recommendation:** 3

**Clarity, Quality, Novelty And Reproducibility:**

The assumption on latent covariate shift for domain adaptation is interesting and makes sense. However, the technical contribution is limited and the experiments are insufficient, which restrict the originality and qulity. The main idea and method are well organized. Important implementation details are missing and the code is not provided, based on which the method and results are not easy to be reproduced.

**Strength And Weaknesses:**

Strengths:
1. The idea of considering latent covariate shift is interesting and the motivation is clearly explained with examples.
2. By combinining difference components, the proposed method achieves superior performance as compared to the compared baselines.


Weaknesses:
1. The latent casual model is naive simply by combining existing disentanglement and casual methods. The insights behind such combination are insufficient.
2. The technical contribution is limited. The different components of the method, such as Gaussian prior, the mutual information and its approximation, entropy regularization, independence enhancement, are all existing techniques.
3. The commonly used datasets for multi-source domain adaptation, such as Digits-five, Office-Home, and DomainNet, are not considered. Due to page limit, I expected to see the results in supplementary material.
4. It is strange that when KL increases from 0.3 to 0.7, the performance of the baselines increase while the proposed method decreases. There is no analysis on the reasons.
Commonly used visualizations for domain adaptation, such as GradCam, are missing. t-SNE is not convincing.
5. The contributions of this paper are not well summarized.
6. Some closely related recent methods on multi-source domain adaptaion are not introduced and compred, such as "Self-paced Supervision for Multi-Source Domain Adaptation", "MADAN: multi-source adversarial domain aggregation network for domain adaptation".
7. The presentation needs to be improved. There are some simple grammar errors, such as "a image"->"an image", "nc play a role"->"nc plays a role". The full names of some abbreviations should be given for the first time use, such as ICA. The format of references, especially the names of conferences and journals, is inconsistent.


**Summary Of The Paper:**

This paper studies multi-source domain adaptation. Instead of learning domain-invariant features to address the conditional shift, this paper assmes latent covariate shift and proposes latent causal model to formulate the data and label generating process. The identifiability of the proposed model is analyzed. By integrating the identifiability with the dependence between nc and y, this paper proposes a method to learn the invariant conditional distribution pD(y|zc). Experiments on both synthetic and real datasets are conducted.

**Summary Of The Review:**

Interesting idea, limited technical contribution, insufficient experiments and analysis and just so-so presentation.

---

> ### Author Response · Authors · 2022-11-18
> **Rebuttal**
>
> __*Q1: The latent casual model is naive simply by combining existing disentanglement and casual methods. The insights behind such combination are insufficient.*__
> Intuitively, for a causal system, there is a corresponding latent noise variable $n_i$ relating to one latent causal variable $z_i$, where $n_i$ are mutually independent. Note that Nonlinear ICA aims to disentanglement independent latent variables $n_i$. So it is natural to leverage the independence to connect latent causal model and nonlinear ICA. In addition, recent progress in the existing disentanglement models (Nonlinear ICA) has shown identifiability results. Therefore, it is also natural to leverage the identifiability results from nonlinear ICA to analyse the identifiability results of latent causal model. Further, the identifiability results of nonlinear ICA is based on an observed variable (e.g., $D$ in Figure 2.). Here $D$ can be regarded as domain index, for which it is straightforward to obtain such observed variable in domain adaptation.
>
> __*Q2: The technical contribution is limited.*__
> Our main contributions can be summarized as follows: 1) Differ from the commonly-used Conditional Shift as shown in Figure 1 (b), which assumes label distribution to be the same across domains, we propose a new problem setting as shown in Figure 1 (c). 2)Leveraging the existing identifiability results of nonlinear ICA, we provide analysis about the identifiability of the proposed latent causal graph in Figure 2, e.g., Proposition 4.1 and Proposition 4.2, which provides guarantee for identifying the latent causal content variable $\mathbf{z}_c$. 3) With the identifiability result of Proposition 4.2, we design a new method for domain adaptation, and show experimental results on simulation and real data. Indeed, we use the existing techniques, e.g, VAE and mutual information, however, these existing techniques are effectively integrated into a framework, based on the guarantee of Proposition 4.2, not heuristic combination. We have summarised our contributions in the new version.
>
> __*Q3:The commonly used datasets for multi-source domain adaptation, such as Digits-five, Office-Home, and DomainNet, are not considered.*__
> We do not consider the commonly used datasets, because for these dataset the label distributions of any two domains are very similar. In this case, these datasets are suitable for domain adaptation with conditional shift as shown in Figure 1 (b). However, these datasets are not appropriate for the proposed setting where the label distribution changes across domains, as shown in Figure (c). Therefore, we resample the original PACS dataset, and also use a new dataset, TerraIncognita. This dataset is commonly used in domain generalization.
>
> __*Q4: It is strange that when KL increases from 0.3 to 0.7, the performance of the baselines increase while the proposed method decreases. There is no analysis on the reasons. Commonly used visualizations for domain adaptation, such as GradCam, are missing.*__
> 1. Generally speaking, larger KL means that domain adaptation on this case is more changeling. However, due to various uncontrollable factors, e.g., the different sample sizes and local optimization, it is not necessarily means that we must have better results on larger KL case than results on smaller KL case, especially in the case where the change of KL is small (e.g., from 0.3 to 0.7). That is, we think the slightly fluctuations of performance are reasonable. Overall, our experimental results verify the proposed method is better than invariant representations based methods, e.g., MCDA, M3DA, LtC-MSDA and T-SVDNet, and the methods, including IRM, IWCDAN and LaCIM, which allows label distribution to change across domains. Such conclusion can be further verified from the results on dataset TerraIncognita.
> 2. T-SNE is not convincing. T-SNE is commonly used for domain adaptation, e.g, Ref. MCDA ((Saito et al., 2018)), M3SDA (Peng et al., 2019),LtC-MSDA (Wang et al., 2020),T-SVDNet (Li et al., 2021).
>
> __*Q5: The contributions of this paper are not well summarized.*__
>  We have added a new paragraph to summarize our contributions in Introduction.
>
> __*Q6: Some closely related recent methods on multi-source domain adaptaion are not introduced and compred, such as "Self-paced Supervision for Multi-Source Domain Adaptation", "MADAN: multi-source adversarial domain aggregation network for domain adaptation".*__
>  We have provided a comparison in Related work for these two works. We note that Ref. "Self-paced Supervision for Multi-Source Domain Adaptation" do not provide their code. We will provide experimental results of Ref. "MADAN: multi-source adversarial domain aggregation network for domain adaptation" in our final version.

---

> > ### Author Response · Authors · 2022-12-04
> > **Could you please provide your further feedback on our response?**
> >
> > Dear Reviewer wpga,
> >
> > We appreciate your time devoted to reviewing the manuscript. We have provided responses to your comments and an updated submission.
> > Could you please check whether they properly addressed your concern? Your feedback would be appreciated. Thank you very much!

---

> > > ### Comment · Reviewer_wpga · 2022-12-06
> > > **further feedback to the authors's response**
> > >
> > > Thank the authors for the rebuttal. I have read both the authors' responses and other reviewers' comments. The authors addressed some of my concerns, such as the used datasets. Besides my own concerns on the limited technical novelty, insufficient analysis, and bad presentation, I also agree with other reviewers on the unclear assumptions (fg87) and unconvincing interpretation of the results (x2cf). There are many things to be improved, so I keep my original rejection rating. Hope the reviewers' comments are helpful to improve this paper and good luck to the authors in another submission.

---

### Official Review · Reviewer_fg87 · 2022-10-24

**Confidence:** 5
**Clarity, Quality, Novelty And Reproducibility:** See above
**Correctness:** 2
**Technical Novelty And Significance:** 3
**Empirical Novelty And Significance:** 2
**Recommendation:** 3

**Strength And Weaknesses:**

Strengths

This work fits in well with a recent line of attacks on domain adaptation that assume an underlying (causal) structure involving latent variables, then define the allowed class of shifts in terms of shifts on these latent variables. Such approaches seem promising for articulating more refined (and possibly even useful) domain shift assumptions. The idea of actually trying to learn the latent variables also seems sound, and relatively underexplored.

Weaknesses

The main problem with this paper is that the main contribution is to give conditions for when the latent-variable modelling approach will work, but then these conditions get omitted!  Proposition 4.1 just references "certain assumptions"! The issue is that all of the promise of this approach comes down to whether or not these certain conditions are actually plausible in real problems. As such, one would expect the bulk of the paper to be dedicated to this question. Instead, the conditions aren't even given.

A secondary---related---weakness is that the treatment of related work fails to make it clear how other approaches to domain adaption relate to the currently proposed one. This is most obvious with respect to latent variable based approaches---e.g., 1,2,3,4 at end of this review. But it's also important with respect to other baselines. For example, if Z_c can be perfectly reconstructed from X, then because Z_c is sufficient for Y under the assumed causal model, we would expect vanilla ERM to work for this problem! It's important to discuss how this, and other distributional approaches, might fail.

Finally, there are some serious problems with clarity. As particular examples,
1. it's unclear what the data setup here is. What qualifies as out of domain? What do we assume access to at training time? The setup seems to be non-standard, but it's not explicitly spelled out
2. How exactly are n and z meant to differ? As latent variables, they are literally indistinguishable in the model in figure 2 (the nodes could be collapsed together without any observable implication)


Some related work on domain adaptation with latent variables following a causal structure:
1. https://proceedings.neurips.cc/paper/2021/hash/a8f12d9486cbcc2fe0cfc5352011ad35-Abstract.html
2. https://arxiv.org/abs/2006.07500
3. https://arxiv.org/abs/2106.00545
4. https://arxiv.org/abs/2208.06987

**Summary Of The Paper:**

This paper considers a domain adaptation setting where the features X and label Y are both influenced by latent features Z, the distribution of which may shift across domains. The idea of the paper is to fit a latent variable model to learn Z, and then use the learned variables as the basis for an invariant predictor. The bulk of the paper is concerned with showing this is indeed possible, which they aim to do by adapting results from non-linear independent component analysis.

**Summary Of The Review:**

Although the core idea of applying non-linear identification to domain adaptation is interesting, this paper is not ready for publication.

---

> ### Author Response · Authors · 2022-11-18
> **Rebuttal**
>
> __*Q1: What are certain assumptions?*__
> The certain assumptions are the assumptions for the identifiability result of nonlinear ICA. We have added a new section for detailed assumptions, see A.3 in the new rebuttal version.
>
> __*Q2: Some related work on domain adaptation, e.g., Refs. [1,2,3,4]*__
> In the original version, we do not discuss these related works, because our work focuses on domain adaptation, which is different from these works (Refs. [1,2,3,4]) in that Refs. [1,2,4] are related to domain generalization (Note that Ref. [4] is still unpublished up to now.), Ref. [3] is related to text classification [3]. Perhaps the closer to our problem setting is domain generalization. In general, because one can not `see' input data $\mathbf{x}$ for domain generalization, obtaining identifiability result in the setting of domain generalization is generally not possible. In contrast, this work provides the identifiability result, providing a principled way to guarantee that the learned predictor can generalize to the target domain. We have added a new paragraph in Related Wok to discuss causality and domain generalization.
>
> __*Q3: it's unclear what the data setup here is. What qualifies as out of domain? What do we assume access to at training time? The setup seems to be non-standard, but it's not explicitly spelled out.*__
> First, to align the following discussion, we consider that 'the data setup here' means the process of resampling the orignal PACS dataset. As we mentioned in experiments, to obtain the situation of LCS on PACS dataset, we filtered the original dataset by re-sampling it to generate three datasets, resulting in different label distributions across domains. In the original PACS, the label distributions for any two domains is very similar (e.g., KL divergence is less than 0.1). So the original PACS data is suitable for domain adaptation with conditional shift as shown in Figure 1 (b), while it is not appropriate for the proposed setting where the label distribution changes across domains, as shown in Figure (c). Therefore, we randomly re-sample the original PACS dataset to provide new PACS dataset where the label distribution changes across domains. We have clarified this point more clearly in  APPENDIX.
>
> __*Q4:How exactly are n and z meant to differ? As latent variables, they are literally indistinguishable in the model in figure 2 (the nodes could be collapsed together without any observable implication).*__
> 1) On model: For a causal system, the latent variables $\mathbf{n}$ represent some unmeasured factors that influence the latent causal variables $\mathbf{z}$. The latent noise variables $\mathbf{n}$ are assumed to be independent, conditional on the observed variable $D$, while there are causal relations among the latent causal variables $\mathbf{z}$, as shown in Figure 2.
>  2) On identifiability: The identifiability result of nonlinear ICA ensures to recover all $n_i$ up to permutation and scaling. From the graph structure in Figure 2, we can see that $y$ depends on $n_c$ only, so that we can measure this dependence ( e.g., by mutual information) to identify which $n_i$ are corresponding to $n_c$. As a result, we can identify $n_c$ and $n_s$. Also, for the graph structure, $z_c$ is only caused by $n_c$, so that we can identifying $z_c$ up to a (nonlinear) transformation. Please see Proposition 4.2 for more details. We can not identify $z_s$, due to the transitivity. Please see Proposition 4.1 for more details.
>
> __*Q5: if zc can be perfectly reconstructed from X, then because zc is sufficient for Y under the assumed causal model, we would expect vanilla ERM to work for this problem.*__
> Proposition 4.2 provides guarantee for identifying $\mathbf{z}_c$. Please note that, the guarantee is ensured under some conditions from nonlinear ICA. Therefore, to ensure that the recovered latent $\mathbf{z}_c$ is consistent with the true one, we need to incorporate these conditions into methods. For ERM, it do not reconstruct the observed data $\mathbf{x}$, which is a necessary condition to recover the true latent $\mathbf{z}_c$.

---

> > ### Author Response · Authors · 2022-12-04
> > **A kind request for further feedback for your concern**
> >
> > Dear Reviewer fg87,
> >
> > Thanks for providing the initial comments. We have provided responses to your main concerns. If there is any other concern, please let us know, and we will immediately respond to that. We understand you are very busy and appreciate your time. Your feedback is valuable to us. Thank you.

---

### Author Response · Authors · 2022-11-19
**General Response**

We thank the reviewers for their valuable feedbacks and time devoted to our work.

Reviewers are positive in topic of the manuscript. Reviewer, fg87, consider that 'The idea of actually trying to learn the latent variables also seems sound, and relatively underexplored.' Reviewer, wpga, think that 'The idea of considering latent covariate shift is interesting and the motivation is clearly explained with examples.' Reviewer, x2cf, consider that 'This paper proposes to solve an important question in multi-source domain adaptation.' and 'the proposed framework shows the potential to solve this challenge.'.

Reviewer fg87, raised a main concern about the detailed assumptions and related, we have addressed it in the individual response.

Reviewers wpga, raised a main concern on the novelty, which are addressed by summarizing our contributions in the new version.

Reviewer x2cf raised a main concern about the details of dataset. We have further clarified these data details in APPENDIX.

We have answered all questions (see more details in the individual responses).

---

### Decision · Program_Chairs · 2023-01-20

**Decision:**

Reject

**Justification For Why Not Higher Score:**

N/A

**Justification For Why Not Lower Score:**

N/A

**Metareview: Summary, Strengths And Weaknesses:**

Although this paper received good appreciations about the idea of latent covariate shift, reviewers' ratings are all below threshold, and the situation is not improved after authors' rebuttal.
Main raised issues concern the insufficient novelty and technical contributions, unclear assumptions under which this approach should work and the positioning wrt the current domain adaptation methods (while also missing to quote some related paper in the state of the art), weak experimental analysis, insufficient to justify the paper claims and, especially, the clarity of the presentation.
Authors provide replies for such remarks, but they did not result all so convincing, and the reviewers maintained their original evaluations.
For these reasons, this paper cannot be accepted to ICLR 2023.

**Summary Of Ac-Reviewer Meeting:**

N/A